# Architecture of the biofilm-associated archaic Chaperone-Usher pilus CupE from *Pseudomonas aeruginosa*

**Jan Böhning[1], Adrian W. Dobbelstein[2◐], Nina Sulkowski[1◐], Kira Eilers[3], Andriko von Kügelgen[1], Abul K. Tarafder[1], Sew-Yeu Peak-Chew[4], Mark Skehel[5], Vikram Alva[2], Alain Filloux[3], Tanmay A. M. Bharat[1,6]***

**1** Sir William Dunn School of Pathology, University of Oxford, Oxford, United Kingdom, **2** Department of Protein Evolution, Max Planck Institute for Biology Tübingen, Tübingen, Germany, **3** Department of Life Sciences, MRC Centre for Molecular Bacteriology and Infection, Imperial College London, London, United Kingdom, **4** Cell Biology Division, MRC Laboratory of Molecular Biology, Francis Crick Avenue, Cambridge, United Kingdom, **5** Proteomics Science Technology Platform, The Francis Crick Institute, London, United Kingdom, **6** Structural Studies Division, MRC Laboratory of Molecular Biology, Francis Crick Avenue, Cambridge, United Kingdom

◐ These authors contributed equally to this work.
* tbharat@mrc-lmb.cam.ac.uk

**Data Availability Statement:** The cryo-EM density map of the CupE pilus generated in this study has been deposited in the Electron Microscopy Databank (EMDB) under the accession number

## Abstract

Chaperone-Usher Pathway (CUP) pili are major adhesins in Gram-negative bacteria, mediating bacterial adherence to biotic and abiotic surfaces. While classical CUP pili have been extensively characterized, little is known about so-called archaic CUP pili, which are phylogenetically widespread and promote biofilm formation by several human pathogens. In this study, we present the electron cryomicroscopy structure of the archaic CupE pilus from the opportunistic human pathogen *Pseudomonas aeruginosa*. We show that CupE1 subunits within the pilus are arranged in a zigzag architecture, containing an N-terminal donor β-strand extending from each subunit into the next, where it is anchored by hydrophobic interactions, with comparatively weaker interactions at the rest of the inter-subunit interface. Imaging CupE pili on the surface of *P. aeruginosa* cells using electron cryotomography shows that CupE pili adopt variable curvatures in response to their environment, which might facilitate their role in promoting cellular attachment. Finally, bioinformatic analysis shows the widespread abundance of *cupE* genes in isolates of *P. aeruginosa* and the co-occurrence of *cupE* with other *cup* clusters, suggesting interdependence of *cup* pili in regulating bacterial adherence within biofilms. Taken together, our study provides insights into the architecture of archaic CUP pili, providing a structural basis for understanding their role in promoting cellular adhesion and biofilm formation in *P. aeruginosa*.

## Author summary

Many bacteria adhere to surfaces or host cells using filamentous structures termed pili that extend from the bacterial cell and anchor them to their target. Previous studies have

EMD-16683. The corresponding atomic coordinates are deposited in the Protein Data Bank (PDB) under accession code 8CIO. The cryo-EM density map of the CupE pilus 111-113$_{AGA}$ mutant is available under the accession number EMD-16686.

**Funding:** This work was supported by the Wellcome Trust (202231/Z/16/Z to TAMB), the Bert L. & N. Kuggie Vallee Foundation (Scholarship to TAMB), the Leverhulme Trust (Philip Leverhulme Prize to TAMB), the Lister Institute of Preventive Medicine (Lister Prize to TAMB), the UKRI Medical Research Council (MR/K501256/1 and MR/N013468/1: Graduate Studentship to JB; MC_UP_1201/31: core funding to TAMB). The funders had no role in study design, data collection and analysis, decision to publish, or preparation of the manuscript.

**Competing interests:** The authors have declared that no competing interests exist.

characterised various Chaperone-Usher Pathway (CUP) pili, which are common in Gram-negative bacteria. However, little is known about the so-called archaic CUP pili, which are the most widespread type. Archaic CUP pili help maintain the architecture of multicellular bacterial aggregates termed biofilms formed by the pathogen *Pseudomonas aeruginosa* and many others. In this study, we present a cryo-EM structure of the archaic CUP pilus CupE from *P. aeruginosa*, providing a structural basis of how the CupE1 protein forms zigzag-shaped, extended pili. By imaging CupE pili on *P. aeruginosa* cells using electron cryotomography, we show that pili can adopt variable long-range curvature, which may help their role in providing cohesion between cells within the biofilm. Furthermore, structural modelling provides insights into the roles of minor pilin subunits encoded within the *cupE* operon. These results will help advance our understanding of bacterial pili structure and function.

## Introduction

Adhesion of bacterial cells to abiotic and biotic surfaces is crucial for the colonization of new environments, including host invasion during infections and biofilm formation [1–5]. Bacterial adhesion is often mediated by proteinaceous, hair-like cell-surface structures known as pili or fimbriae [6,7]. Pili are assembled by repeated interactions of monomeric protein subunits, resulting in a filamentous structure that protrudes from the cell surface to anchor the cell to substrates [8].

Chaperone-Usher Pathway (CUP) pili are among the most widely distributed and best-characterized adhesins in Gram-negative bacteria [9]. CUP systems are typically encoded as single operons and consist of at least three components: a major pilin subunit, a periplasmic chaperone that stabilizes the pilin prior to assembly, and an outer membrane (OM) usher pore protein responsible for translocation and assembly of the pilin [10–13]. Frequently, CUP operons encode additional components, including adhesins that decorate the tip of the pilus distal to the OM [14], regulatory proteins, minor pilin subunits, and additional chaperones. CUP pilin subunits generally consist of an incomplete immunoglobulin-like (Ig-like) β-sandwich fold that lacks the final antiparallel β-strand, but contains an additional N-terminal β-strand extending away from the subunit core [15]. In the assembled pilus, each pilin subunit provides its N-terminal β-strand to the following subunit to complete its Ig-like fold in a process termed donor-strand complementation [16]. Prior to assembly, the missing β-strand within the fold is donated by the chaperone protein [15]. A tip adhesin subunit typically mediates the adhesive function of the pilus, often capping the pilus and mediating specific interactions with host receptors or other abiotic molecules [17–19]. Additional pilin subunits encoded by many CUP operons typically fulfil a specialized structural role within the pilus [20–22] or have a role in terminating pilus assembly [23].

CUP pili are phylogenetically divided into three classes: classical, alternative, and archaic [6]. The best-characterized CUP pili belong to the classical type and include the Fim (Type 1 pilus) [17,24] and Pap (P pilus) [21,25] systems of *Escherichia coli*, which form stiff, tubular structures important for pathogenicity [24,26]. However, available structural information on non-classical CUP pili is scarce, being limited to a crystal structure of a pilin of the archaic Csu pilus [27]. Moreover, there have been no high-resolution studies of CUP pili in their cellular environment. Archaic CUP pili, the phylogenetically oldest class, are widespread in all proteobacteria, cyanobacteria, and even in some extremophilic phyla such as *Deinococcota* [6]. The best-characterized archaic CUP pili include the CupE system in *P. aeruginosa* [28,29] and the

Csu system in *Acinetobacter baumannii* [19,27,30]. Both of these examples are from bacterial species belonging to the ESKAPE class of multidrug-resistant pathogens for which new antibiotics must urgently be developed [31], and targeting antimicrobials against pilins has been suggested as a potential therapeutic avenue [32]. Archaic CUP pili are crucial for the formation of biofilms [28,30], which play an important role in persistent and chronic infections [33,34].

The archaic CupE pilus of *P. aeruginosa* is thought to have been acquired by horizontal gene transfer and evolved independently of other *cup* clusters in the genome (*cupA-D*), all of which belong to the classical type of CUP systems [28]. The *cupE* gene cluster encodes one major pilin subunit (CupE1) that is the main component of the pilus, two minor pilin subunits (CupE2, CupE3) whose arrangement or function within the pilus is unknown, a chaperone (CupE4), an usher (CupE5), and a tip adhesin protein (CupE6). The *cupE* operon is activated by a two-component system, PprA-PprB, and plays an important role in micro- and macrocolony formation and in maintaining the three-dimensional shape of the biofilm [28,35]. The expression of all CUP pili in *P. aeruginosa* is tightly regulated [36], and the CupE pilus is expressed as part of an exopolysaccharide-independent adhesive signature together with the type I secretion system-dependent adhesin BapA, Type IVb pili and extracellular DNA [35]. Hence, its function appears distinct from the other CUP systems in *P. aeruginosa* (CupA-D), which appear to be part of a different adhesive signature [36–38].

As detailed above, comparatively less is known about the structure and architecture of archaic CUP pili than their classical counterparts, despite their presence in a wide range of species [6]. To bridge this critical knowledge gap, we purified CupE pili that were overproduced in *P. aeruginosa* cells by deleting the *mvaT* gene, which encodes a CUP repressor, as well as PA2133, which encodes a phosphodiesterase in the *cupA* operon. We used electron cryomicroscopy (cryo-EM) and cryotomography (cryo-ET) on natively assembled CupE pili *in vitro* and *in situ* to elucidate their structural and architectural properties. We combined our structural and imaging experiments with bioinformatics, revealing important insights into the interdependence and role of CUP pili in biofilm formation of *P. aeruginosa*.

## Results

### Purification of natively assembled CupE pili

To study the role of CupE pili in *P. aeruginosa*, we engineered a strain with increased expression of CupE to facilitate isolation and structural analysis. Therefore, we used a previously described methodology of studying strains in which other common cell surface filaments such as Type IV pili and flagella had been deleted [37], which are otherwise abundant in pilus preparations. In addition, the gene encoding the MvaT repressor was deleted, as this had previously been shown to result in overexpression of CUP pili in *P. aeruginosa* [36]. Thus, all further experiments were performed using cells with a Δ*pilA* Δ*fliC* Δ*mvaT* background. During the bioinformatic analysis of *cup* operons in *P. aeruginosa*, we noted that the classical-type *cupA* operon also encodes a phosphodiesterase (PA2133, hereafter referred to as *cupA6*). As suggested by a previous study [39], we reasoned that deletion of this gene may increase cellular cyclic di-guanylate (c-di-GMP) levels, a biofilm master regulator, and thus further increase the expression of CUP pili. Indeed, deletion of the *cupA6* phosphodiesterase resulted in increased expression of thin cell surface filaments, as quantified via negative stain electron microscopy, with 188 fibres detected on cells of a Δ*cupA6* strain versus 60 fibres detected in a control over 30 micrographs acquired randomly (S1A Fig and Methods). This strain with increased expression of surface filaments was used to obtain preparations of purified filaments by shearing from the surface of cells scraped from plates (Methods). In the purified specimen, we observed long, curved filaments on cryo-EM grids (Fig 1A), with a different morphology than previously

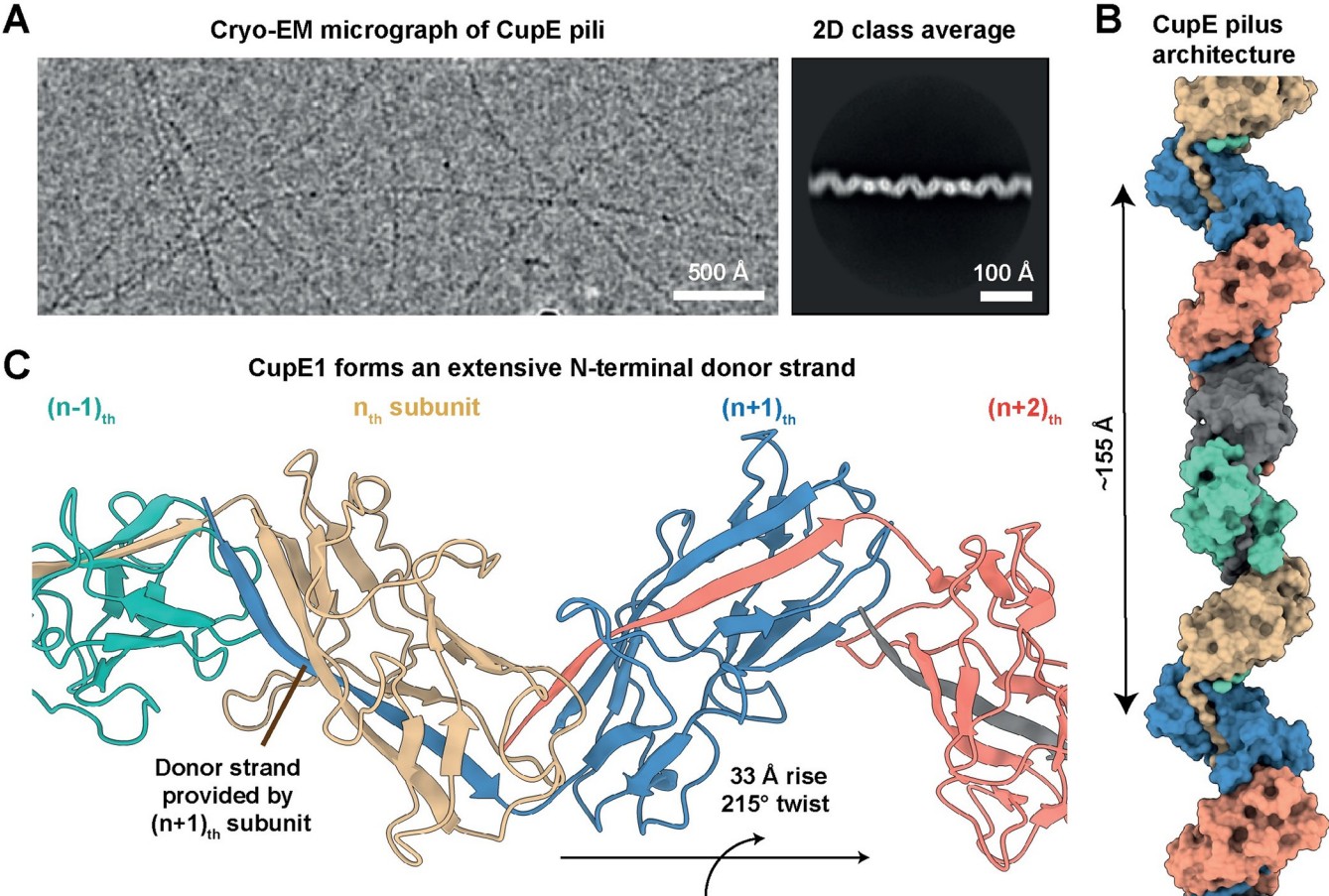

**Fig 1. CupE1 subunits within the CupE pilus are arranged in a zigzag architecture. (A)** Raw micrograph and 2D class average of CupE pili show zigzag-shaped filaments. **(B-C)** A 3.5 Å resolution cryo-EM structure of the CupE pilus reveals how CupE1 subunits are arranged in a zigzag pattern, with the donor strand of the $(n+1)_{th}$ subunit enveloped by the incomplete Ig-like fold of each $n_{th}$ subunit. Five CupE1 subunits form a longer repeat of ~155 Å.

observed for classical-type CUP filaments [25], suggesting that these pili may correspond to archaic-type CupE pili rather than classical-type, tubular-shaped CupA-D pili. Indeed, mass spectrometric peptide fingerprinting confirmed that the sample contained CupE1 protein. These observations suggest that the combined deletion of genes encoding the MvaT repressor and the phosphodiesterase CupA6 caused increased expression of CupE filaments. To further verify that the observed filaments are CupE pili, we deleted *cupE1-E2* in the Δ*cupA6* background and found that the long, curved filaments with an apparent diameter of 60 Å were no longer present in fractions sheared from the cell surface (S1B and S1C Fig).

## Atomic structure of CupE pili using cryo-EM helical reconstruction

After confirming the presence of natively assembled CupE pili in our preparation, we proceeded with cryo-EM analysis. Cryo-EM micrographs and 2D classes of CupE showed a zigzag appearance of the pili (Fig 1A), with a diameter of approximately 60 Å. Interestingly, some pili spontaneously associated into large mesh-like bundles (S1D Fig). After deducing the symmetry of the pilus from two-dimensional class averages of single filaments (Fig 1A), we performed helical reconstruction and obtained a 3.5 Å-resolution cryo-EM density, from which an atomic model of the CupE pilus could be built (Figs 1B, 1C and S2 and S1 Table and S1 Movie). The

atomic model reveals an arrangement of CupE1 subunits in a zigzag architecture (215˚ right-handed rotation per subunit, Fig 1B and 1C). In agreement with classical CUP proteins, the N-terminal β1-strand of each $(n+1)_{th}$ subunit completes the Ig-like fold of the $n_{th}$ subunit (Fig 2A and 2B). This donor strand interacts with the $n_{th}$ subunit through both β-sheet hydrogen bonding as well as hydrophobic interactions, filling a hydrophobic groove in the subunit core (Figs 2C and S3). The structure also shows two key cysteine residues (C41-C85) forming a disulfide bridge within a β-sheet from which the donor strand extends (Fig 2D). Bioinformatic analysis shows that these cysteines are highly conserved (S4 Fig), suggesting they may play an important role in maintaining subunit stability.

The 13-residue donor strand of the $(n+1)_{th}$ subunit represents the majority of the interaction with the $n_{th}$ subunit. The donor strand is connected to the core of each subunit through a Glycine-Alanine-Glycine (GAG) sequence (Fig 2D). The interface between the Ig-like domains is relatively small in comparison to the extensive donor-strand interactions, and is mainly maintained by the interaction of a loop with the next subunit, with two interacting valine residues V69 and V48 at the core of the interaction (Fig 2D). In a recent publication on the related Csu pilus, interactions between zigzag-arranged subunits within an archaic CUP pilus were termed 'clinch' interactions [40]. While the subunit arrangement is overall similar in the CupE pilus, there are significantly fewer interacting residues between subsequent subunits in the CupE pilus compared to the Csu pilus, despite the high structural and sequence similarity between the pilins (RMSD = 1.55 Å; S4 and S5 Figs). Thus, the CupE pilus shows a relaxed arrangement of subunits compared to the tight packing of the Csu pilus, resulting in a higher rise per subunit than in the Csu pilus (S5B Fig). The relaxed subunit arrangement suggests the possibility that the CupE pilus may be more flexible than the Csu pilus, which has been described as laterally stiff [40]. Indeed, increased lateral curvature within CupE pili was observed in a subset of two-dimensional class averages (S3B Fig), supporting some degree of flexibility around the subunit-subunit interface that allows the orientation of subunits to vary.

Visual inspection of the CupE pilus along the helical axis (S3C Fig) showed a serine and threonine-rich loop (`TTTTSST`) extending outward from the helical axis of the CupE pilus, constituting the part of the subunit that is most exposed to the environment (S3C Fig). Sequence alignment of archaic pilin subunits shows that this unusual serine and threonine-rich sequence is unique to *P. aeruginosa* (S4 Fig). In the EM map, we noticed extra density near the loop that could not be attributed to the atomic model of the CupE1 protein subunit and that we suspected might correspond to a post-translational modification. Mutation of three threonine residues within the loop to Alanine-Glycine-Alanine, followed by cryo-EM structure determination, resulted in a 4.1 Å resolution map with a reduced density in this region (S3D and S3E Fig and S1 Table), confirmed by calculating difference maps with the wild-type pilus structure (S3F Fig). While disassembly into monomers followed by intact mass spectrometric analysis allowed us to detect a molecular species 206 Da larger than the expected mass of the CupE1 monomer (S3G Fig), predominantly unmodified CupE1 protein was observed. Thus, the chemical identity of the extra density near the threonine-rich loop cannot be ascertained at this stage.

## Imaging of CupE pili *in situ* on the surface of *P. aeruginosa* cells

CupE pili have been shown to promote the mature architecture and mushroom shape of *P. aeruginosa* biofilms [28]. To find out how CupE pili support cohesion between cells within the biofilm, we imaged CupE pili directly on the *P. aeruginosa* cell surface. To this end, we deposited cells from colonies of the Δ*cupA6* strain used above for obtaining pure CupE preparations onto grids for electron cryotomography (cryo-ET) imaging. In the resulting cellular electron

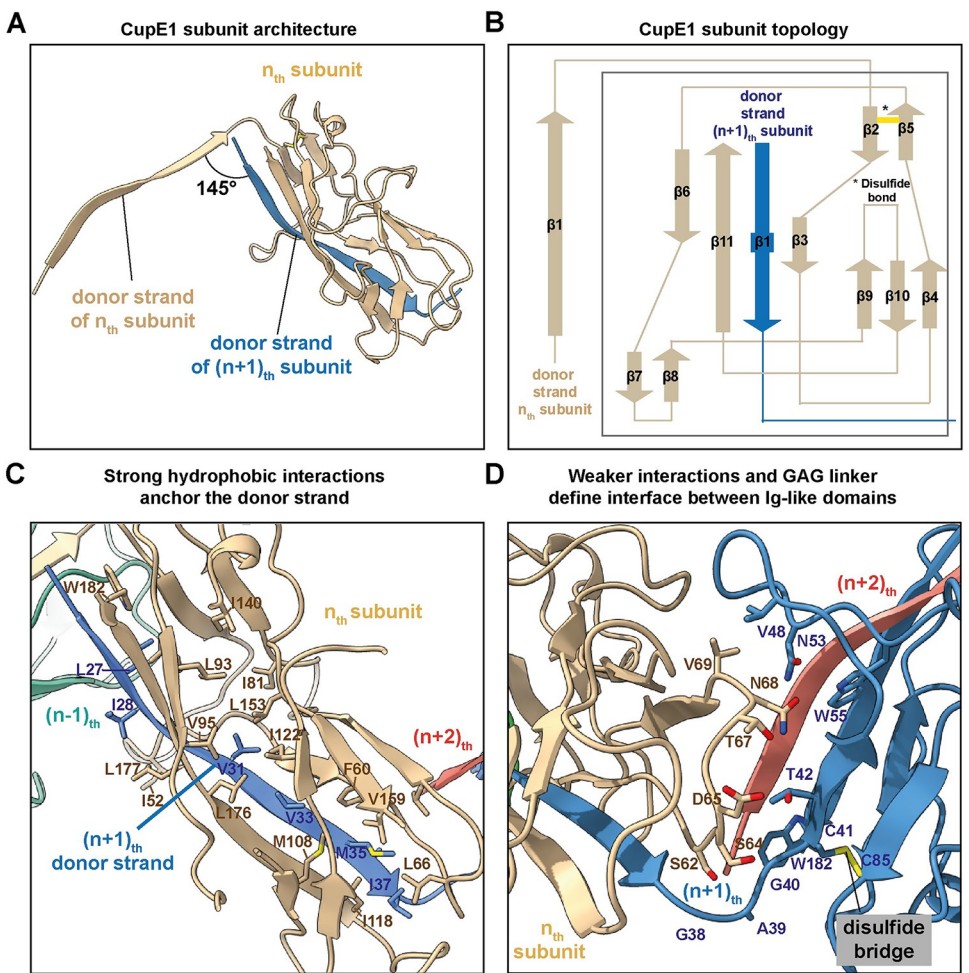

**Fig 2. Architecture of the CupE1 subunit within the CupE pilus. (A)** The donor strand from the $(n+1)_{th}$ subunit completes a β-sheet in the $n_{th}$ subunit by providing a 13-residue β-strand. **(B)** Subunit topology demonstrating β-sheet architecture of the CupE1 subunit. A yellow line denotes a disulfide bridge. **(C)** Extensive hydrophobic interactions anchor the donor strand of the $(n+1)_{th}$ subunit into the Ig-like fold of the $n_{th}$ subunit. **(D)** Compared to the extensive interactions of the donor strand with the complemented subunit, the globular subunit:subunit interface between Ig-like folds shows few strong interactions at the interface. A disulfide bridge is observed in each CupE1 subunit (marked).

cryotomograms (Fig 3), we observed CupE pili with the same characteristic zigzag architecture seen in *vitro* extending from the cell surface (Figs 3A, 3B and S6). To confirm that these were the same CupE pili, we extracted subtomograms along the length of the pili in tomograms and performed subtomogram averaging [41]. Our subtomogram averaging structure of the cell surface filaments, produced from an unbiased cylindrical reference, recaptures the CupE pilus' zigzag appearance (S6 Fig), which was observed even without symmetrization, validating the identity of these filaments as CupE pili. Interestingly, CupE pili attached to the cell could adopt variable curvatures *in situ* (Figs 3, S6C and S6D and S2 Movie). In small *P. aeruginosa* cell clusters, pili were repeatedly found to extend from and fold back onto the originating cells (Fig 3C and 3D). We estimated the local curvature of CupE pili in cellular tomograms and discovered local deviation from the helical axis of up to 15˚ per subunit (S6E Fig). This observed ability of CupE pili to locally adopt higher curvature contrasts with classical, tubular CUP pili, which are comparatively rigid assemblies [25]. At the same time, however, it also differs from γ-

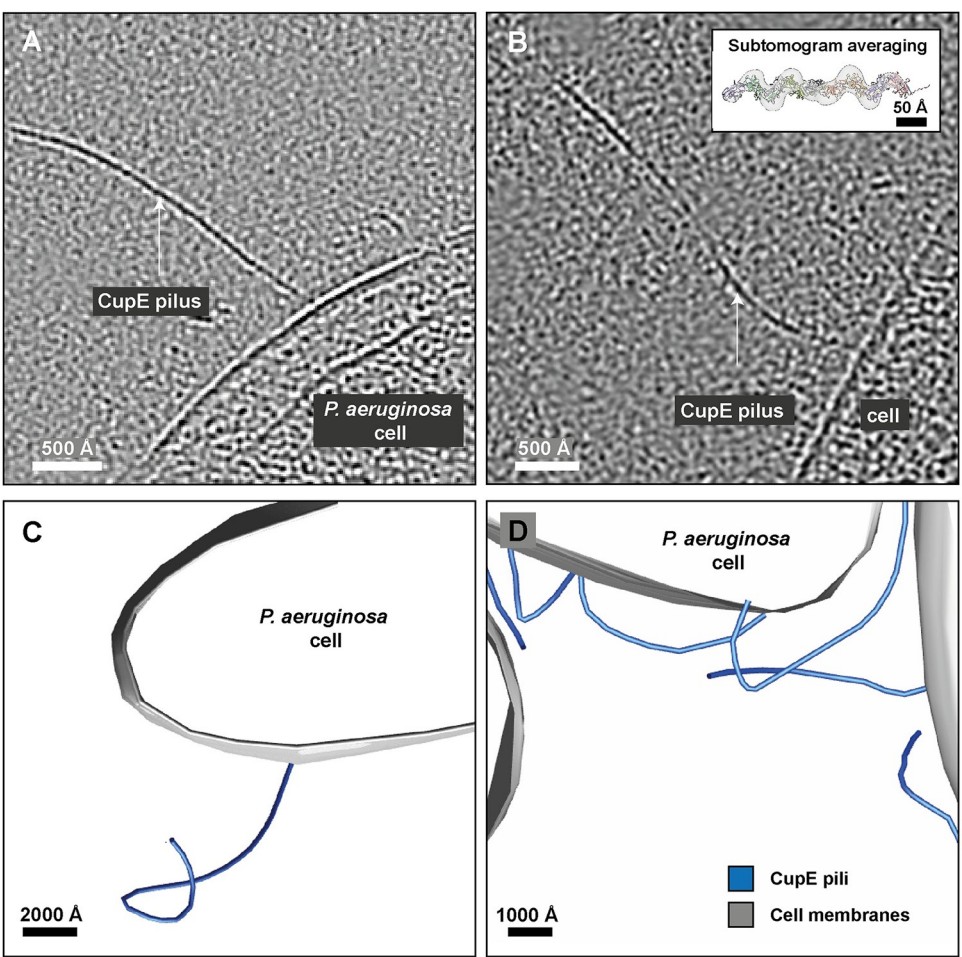

**Fig 3. Cryo-ET imaging of CupE pili on *P. aeruginosa* cells. (A-B)** Tomographic slice of cells expressing CupE pili. CupE pili adopt significant curvature on cells. The pilus can be seen going in and out of plane in (B). Inset: Atomic model of CupE fitted in the subtomogram averaging map produced from cellular cryo-ET data. **(C-D)** Segmentation of cell membranes and CupE pili illustrates how CupE pili adopt variable curvatures. See also S2 Movie.

clade linear Chaperone-Usher fibres, such as the Caf1 and Saf pilus that were found to have extremely low persistence lengths and even higher angles between subunits [42,43]. Adopting curvature may be required to support efficient interactions of the pilus in the extracellular matrix of the biofilm, allowing the cells to embed themselves in a matrix rich in filamentous molecules. Here, some flexibility of the pilus may help promoting cell-cell adhesion within the biofilm.

## Structure- and sequence-based bioinformatic analysis of CupE pili

To place the structural and imaging data on CupE pili described above into the context of *P. aeruginosa* biofilm formation and CUP pili regulation, we performed computational analyses, facilitated by recent developments in methods for protein structure prediction [44] and sequence analysis. While our cryo-EM structure is in agreement with previous studies that showed CupE1 to be the major CupE pilin subunit [28], the *cupE* operon also encodes two additional predicted pilin subunits, CupE2 and CupE3, which exhibit high pairwise sequence similarity to CupE1, 39% and 30%, respectively (S4 Fig). To model the oligomers formed by the CupE2 and CupE3 subunits, we used AlphaFold-Multimer, which has been shown to

predict the structures of monomeric and multimeric proteins with atomic-level accuracy [45]. Our modelling suggests that CupE2 and CupE3 can form donor-strand-exchanged filaments similar to CupE1, being able to form homofilaments as well as heterofilaments with CupE1 (S7A and S7B Fig). As a further validation of our modeling, we found that AlphaFold2 correctly predicted the CupE1 subunit arrangement consistent with our cryo-EM data (S7 and S8 Figs), with the subunit-subunit interface differing only slightly to our experimental structure. Finally, we also used AlphaFold2 to predict a structural model of the putative complex formed between CupE1 and the tip adhesin subunit CupE6. In the obtained model, CupE6, like the tip adhesin CsuE of *A. baumannii* [19], is composed of two Ig-like folds. While the C-terminal domain is incomplete and caps the CupE1 filament by accepting a donor strand from CupE1, the distal end of the N-terminal fold contains a highly hydrophobic surface patch that likely plays a role in adhesion to target substrates (S7C Fig).

To investigate the conservation and co-occurrence of *cup* gene clusters in isolates of *P. aeruginosa* at the genomic level, we performed a comprehensive sequence search of the *Pseudomonas* Genome Database [46] and the NCBI RefSeq database. We searched for the occurrence of CupA-D and CupE systems with the usher as the query sequence (CupA3-CupD3 and CupE5), as the usher is the most conserved protein among the genes encoded by these clusters [6]. We detected complete *cupE* gene clusters with conserved gene order in 228 out of 233 strains of *P. aeruginosa* in the *Pseudomonas* Genome Database, showing the widespread occurrence of this archaic CUP cluster. Notably, *cupE* is absent in the strain PA7, a highly divergent isolate of *P. aeruginosa* [47]. Of the four classical *cup* clusters *in P. aeruginosa* (CupA-D), we identified the *cupB* cluster in all 233 strains, whereas the *cupA* and *cupC* gene clusters were absent in only a few strains (S5 Table and S1 Data). In contrast, the *cupD* gene cluster, previously described to be located on the pathogenicity island PAPI-1 [38,48], was found only in 11 strains. Interestingly, both the *cupA* and the *cupE* clusters are missing in most of the strains containing the *cupD* cluster. Given the high sequence similarity between the corresponding protein subunits of the CupA and CupD systems (>65% pairwise sequence identity), we speculate they serve very similar functions and are, therefore, often mutually exclusive. Taken together, three different classical *cup* gene clusters and one archaic gene cluster were found to be widespread in *P. aeruginosa* strains.

The success of our strategy to upregulate CupE pili by deleting a phosphodiesterase-encoding gene in the *cupA* operon (*cupA6)* suggests that CUP pili expression may be interdependent on other CUP genes, suggesting possible co-regulation. This agrees with the findings of our bioinformatics analysis, as CupA and CupE co-occur in almost all strains of *P. aeruginosa*. Notably, the phosphodiesterase CupA6 is encoded after the chaperone CupA5 with a sequence overlap of four nucleotides in every strain containing CupA (Fig 4A), indicating a likely translational coupling between them [49].

Outside *P. aeruginosa*, *cup* gene clusters are more divergent and do not share a conserved gene order, exhibiting missing, additional, or swapped genes. In other bacteria of the genus *Pseudomonas*, *cupE*-like gene clusters are more widespread than *cupA-cupD*, and the co-occurrence of *cupA* and *cupE* is rare (Fig 4); for example, while *P. putida* NBRC_14164 lacks *cupA*, *P. fluorescens* ATCC_13525 contains both *cupA* and *cupE*. The co-occurrence of *cupE*-like and *cupA*-like gene clusters was also identified in some species of other gammaproteobacteria (e.g., *Yersinia pestis*, *A. baumannii*) as well as betaproteobacteria (e.g., *Burkholderia pseudomallei*) (Fig 4B).

## Discussion

In this study, we present the structure of an archaic CUP pilus, revealing a zigzag architecture with extensive interactions between the donor strand and the complemented subunit. The

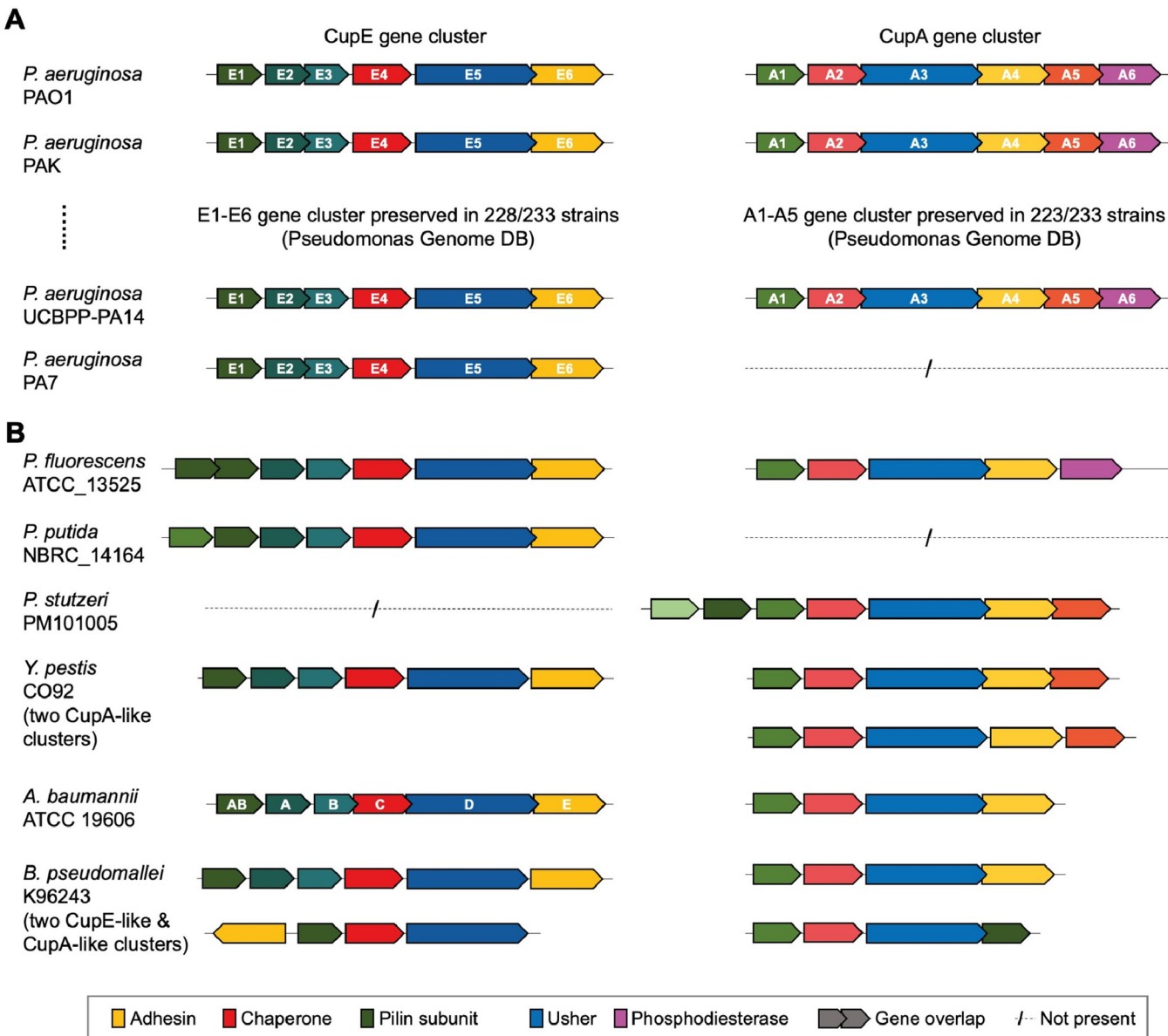

**Fig 4. Co-occurrence of *cupA* and *cupE* gene clusters.** (A) Bioinformatic analysis reveals co-occurrence of the *cupA* and *cupE* gene clusters with preserved gene order in most *P. aeruginosa* strains. Overlapping sequences of different CUP genes indicate possible translational coupling; of the usher gene *cupE5* to the adhesive tip subunit gene *cupE6*, of the two minor pilin subunits *cupE2* and *cupE3*, and of all genes *cupA2-cupA6*. A phosphodiesterase (CupA6) is encoded after *cupA5* in every strain in which *cupA* occurs. (B) Outside of *P. aeruginosa*, examples of co-occurrence can be found in pathogenic proteobacteria such as *P. fluorescens*, *Y. pestis*, *A. baumannii*, and *B. pseudomallei*. Gene order in CUP clusters outside *P. aeruginosa* is not preserved; e.g. in *P. fluorescens* ATCC_13525, an additional pilin is encoded, and in *B. pseudomallei* K96243, the *cupA6*-like gene is encoded on the opposite strand. Accession data and gene loci are provided in S5 Table and supplementary file S1 Data.

subunit interface between the globular part of the Ig-like domains has significantly fewer inter-subunit contacts, and imaging of CupE pili *in situ* shows that they can adopt varied curvatures, which may aid in interactions with the biofilm matrix. This shows that the CupE pilus is not a stiff, tubular assembly like other classical rod-shaped CUP pili such as the P pilus [25], but it is also not highly flexible like the Saf pilus or other FGL-assembled CUP fimbriae [42,43]. The difference in lateral flexibility between the CupE pilus and the closely-related, but

stiff, Csu pilus from *A. baumannii* can be explained by the number of interacting residues at the inter-subunit interface: The Csu pilus, which was characterised in a recent study by overexpression of the corresponding *csu* operon in *E. coli* [40] has increased subunit interactions within the pilus as a result of denser packing compared with CupE (S5 Fig).

Interestingly, it was found in optical tweezer experiments that the Csu pilus can be extended to almost twice its length along the helical axis by breaking of the subunit interface contacts [40]. Given that CupE shares a similar architecture, it seems likely that it shares this super-elasticity, suggesting it may be a common feature of archaic CUP pili. We propose that the lateral curvature of the CupE pilus observed in the cellular environment may stem from the same properties, i.e., the result of breaking some or all non-donor-strand subunit interface contacts.

AlphaFold structure predictions for minor CupE subunits CupE2 and CupE3 suggest that they can also form filaments through donor-strand exchange. While both homofilaments and heterofilaments with CupE1 are predicted to be possible (S7 Fig), only CupE1 homofilaments appear to form the more compact zigzag architecture. However, it is unclear whether the minor pilins CupE2 and CupE3 form homofilaments at certain sections of the pilus, similar to the tips in the classical Type 1 and P pili [9], or whether they are sub-stoichiometrically embedded between CupE1 subunits. Other possibilities are that they functionalize pili or fulfil an undetermined function in pilus assembly. For example, in the Pap system, the PapH subunit terminates pilus assembly [23,50]; whether an analogous protein exists in archaic CUP pili remains to be determined. Structural prediction of the CupE6 adhesin tip subunit also suggests that, in agreement with previous analyses on the Csu system [19], CupE6 contains the same subunit fold and hydrophobic surface at the tip that could interact with other hydrophobic components, thus supporting adhesion. The chemical nature of the substrate of the CupE6 pilus tips in biofilms remains enigmatic, and it is unclear how the highly hydrophobic pilus tip would be stabilized during pilus assembly, presenting an exciting direction for future inquiries.

CupE and Csu, the two archetypal archaic CUP pili systems, are both involved in promoting biofilm formation [28,30]. While we study CupE pili on cells, an intrinsic limitation of our system is that single cells expressing CupE do not fully recapture the molecular sociology and crowding conditions within intact biofilms. The interaction partners of the pilus are hence unknown and warrant the focus of future imaging efforts. Since we observed isolated CupE pili forming regular mesh-like arrays in cryo-EM images (S1D Fig), this might indicate that lateral interactions of pili may occur in the crowded conditions of the biofilm matrix, similar to other biofilm matrix fibre systems [51–53]. Further studies on native cellular systems–i.e., biofilms—will be required to determine the exact mode of interaction of CupE pili with other extracellular matrix components and also with cells.

Our study finds that deletion of the gene encoding the phosphodiesterase CupA6 in the *cupA* operon, results in increased expression of the *cupE* operon. A likely explanation for this is that the *cupA* operon negatively regulates cyclic di-GMP levels through CupA6, and that deletion of CupA6 causes higher cyclic di-GMP levels, which in turn causes *cupE* expression. Indeed, we find that *cupA* and *cupE* gene clusters mostly co-occur in *P. aeruginosa* isolates, suggesting both fulfil distinct functions during biofilm development and that their expression may be co-regulated and interdependent. Numerous CUP systems and other adhesins have been identified in *P. aeruginosa*, and many have been found to be of general importance for biofilm formation. This prompts the overarching question: why does *P. aeruginosa* have an entire arsenal of different adhesins? Are some adhesin systems co-operative, or are they expressed only under specific conditions—and if so, when? Answering these questions in future studies will greatly improve our understanding of adhesion, biofilm formation, and pathogenicity of *P. aeruginosa* specifically and Gram-negative bacteria in general.

## Materials and methods

### Construction of *P. aeruginosa* mutants

*P. aeruginosa* deletion mutants were created as described previously utilising the suicide plasmid pKNG101 [54]. Briefly, to engineer gene deletions in the PAO1 strain, 500 bp DNA fragments of the 5' (upstream) and 3' (downstream) ends of the gene of interest were obtained by PCR using PAO1 chromosomal DNA as a template. The upstream fragment was amplified with the oligonucleotides P1 and P2 while the downstream fragment was amplified using P3 and P4 (S2 Table). A third PCR step using P1 and P4 resulted in a DNA fragment with the flanking region of the gene of interest. The gene fragment was then cloned into pCR-BluntII-TOPO (Invitrogen), the sequence confirmed and sub-cloned into the pKNG101 suicide vector (S3 Table). The pKNG-derivatives were maintained in *E. coli* strain CC118λpir (for strain descriptions, see S4 Table) and conjugated into PAO1 using *E. coli* 1047 harbouring the conjugative plasmid pRK2013. pKNG101 was conjugated into *P. aeruginosa* as described in [54]. After homologous recombination, colonies were streaked onto agar containing 20% (w/v) sucrose and grown at room temperature for 48 hours to select for colonies that have lost the plasmid backbone. Gene deletions were verified by PCR using external primers P5 and P6 (S2 Table).

Chromosomal substitution of TTTTSST to AGATSST in CupE1 was achieved by reintroduction of the previously deleted *cupE1-2* fragment into PAO1 Δ*pilA* Δ*fliC* Δ*mvaT* Δ*cupA6* Δ*cupE1-2*. In brief, the *cupE1-2* DNA fragment was amplified using primers called Δ*cupE1-2* P1 and P4 (S2 Table), subcloned into pCR-BluntII-TOPO (Invitrogen) and subjected to site directed mutagenesis using CupE1-AGATSST Fw and Rev primers (S2 Table). After the sequence was confirmed, the fragment was cloned into pKNG101 and conjugation was conducted as described above. Mutation was verified by PCR using external primers Δ*cupE1-2* P5 and P6 and clones with *cupE1-2* fragment reintroduced were sequenced.

### Isolation of CupE pili

An overnight culture of *P. aeruginosa* PAO1 Δ*mvaT* Δ*cupA6* Δ*pilA* Δ*fliC*, grown in lysogeny broth (LB) medium at 37˚C with agitation at 180 revolutions per minute (rpm), was used to plate lawns on agar plates, and incubated overnight at 37˚C. Bacterial lawns were scraped and resuspended in 1x phosphate-buffered saline (PBS). The resulting suspension was vortexed for 90 seconds to promote dissociation of pili from the cell surface. Cells were then centrifuged at 4,500 relative centrifugal force (rcf) for 20 minutes, and the supernatant centrifuged again 3–4 times at 16,000 rcf to remove remaining cells and cellular debris. 500 mM NaCl and 3% (w/v) PEG-6000 were added to the supernatant, and pili were precipitated on ice for 1 hour. Precipitated pili were collected via centrifugation for 30 minutes at 16,000 rcf. For cryo-EM, precipitated pellets were combined and precipitated again in the same manner and resuspended in 1x PBS to produce the final product.

### Negative stain electron microscopy

2.5 µl of sample was applied to a glow-discharged carbon support grid (TAAB), blotted, washed three times with water, and stained using three 20 µl drops of 2% (w/v) uranyl acetate and allowed to air-dry. Negatively stained grids were imaged on a Tecnai T12 microscope.

### Cryo-EM and cryo-ET sample preparation

For cryo-EM grid preparation of purified CupE pili, 2.5 µl of the sample was applied to a freshly glow-discharged Quantifoil R 2/2 Cu/Rh 200 mesh grid and plunge-frozen into liquid ethane using a Vitrobot Mark IV (ThermoFisher) at 100% humidity at an ambient

temperature of 10°C. For tomography sample preparation of PAO1 Δ*pilA* Δ*fliC* Δ*mvaT* Δ*cupA6*, a bacterial lawn from an overnight LB agar plate incubated at 37°C without antibiotics was resuspended in PBS, and 10 nm Protein-A-gold beads (CMC Utrecht) were added as fiducials prior to plunge-freezing.

### Cryo-EM and cryo-ET data collection

Cryo-EM data was collected in a Titan Krios G3 microscope (ThermoFisher) operating at an acceleration voltage of 300 kV, fitted with a Quantum energy filter (slit width 20 eV) and a K3 direct electron detector (Gatan). Images were collected in super-resolution counting mode using a physical pixel size of 1.092 Å/pixel for helical reconstruction of CupE pili and 3.489 Å/pixel for cellular tomography data. For helical reconstruction of CupE, movies were collected as 40 frames, with a total dose of 45–46 electrons/Å$^2$, using a range of defoci between -1 and -2.5 μm. For the wild-type CupE pilus dataset, 11,584 movies were collected; for the 111-113$_{AGA}$ dataset, 4,665 movies were collected. Cryo-ET tilt series of PAO1 Δ*pilA* Δ*fliC* Δ*mvAT* Δ*cupA6* cells were collected using a dose-symmetric tilt scheme as implemented in SerialEM [55], with a total dose of 121 electrons/Å$^2$ per tilt series and defoci of -8 to -10 μm, and with ±60° tilts of the specimen stage at 1° tilt increments.

### Cryo-EM processing

Helical reconstruction of CupE pili was performed in RELION 3.1 [56–58]. Movies were motion-corrected and Fourier-cropped using the RELION 3.1 implementation of MotionCor2 [59], and CTF parameters were estimated using CTFFIND4 [60]. Initial helical symmetry of CupE pili was estimated through indexing of layer lines and counting the number of visible subunits along the pilus. Three-dimensional classification was used to identify a subset of particles that supported refinement to 3.5 Å resolution. For final refinement, CTF multiplication was used for the final polished set of particles [57,61,62]. Symmetry searches were used during reconstruction, resulting in a final rise of 33.12 Å and a right-handed twist per subunit of 214.56°. Resolution was estimated using the gold-standard Fourier Shell Correlation (FSC) method as implemented in RELION 3.1. Local resolution measurements were also performed using RELION 3.1.

### Model building and refinement

Manual model building of the CupE1 subunit was performed in Coot [63] as follows. A homology model based on the structure of *A. baumannii* CsuA/B (RCSB 6FQA) was calculated using MODELLER [64] and this homology model was fit into the cryo-EM density as a rigid body. Residues of the homology model that were inconsistent with the density, including the N-terminal donor strand, were deleted and manually rebuilt. The initially built model was subjected to real-space refinement against the cryo-EM map within the Phenix package [65,66]. Five subunits of CupE1 were built and used for final refinement. Non-crystallographic symmetry between individual CupE1 subunits was applied for all refinement runs. Model validation including map-vs-model resolution estimation was performed in Phenix.

### Tomogram reconstruction

Tilt series alignment via tracking of gold fiducials was performed using the etomo package as implemented in IMOD [67]. Tomograms were reconstructed with WBP in IMOD or SIRT in Tomo3D [68,69]. Deconvolution of tomograms using the tom_deconv.m script [70] was performed for visualisation purposes.

## Subtomogram averaging

Subtomogram averaging of pili on cells was performed in RELION 4 [41], employing helical reconstruction [56]. A cylindrical reference was used to avoid bias. Helical symmetry was applied to enhance the signal during particle alignment. The map presented in Figs 3 and S6 is unsymmetrized.

## Data visualisation and quantification

Cryo-EM images were visualized in IMOD. Fiji [71] was used for bandpass and Gaussian filtering, followed by automatic contrast adjustment. Atomic structures and tomographic data were displayed in ChimeraX [72]. Segmentation of tomograms was performed manually in IMOD. Quantification of cell surface filaments in the Δ*cupA6* mutant was performed through manual annotation in 30 randomly acquired negative stain EM images targeted on cells located at low magnification. Atomic models are shown in perspective view, except for S3C Fig, which is shown in orthographic view. Hydrophobic surfaces were calculated in ChimeraX using the in-built *mlp* function. Difference maps were calculated using EMDA [73] with maps lowpass-filtered to the same resolution (4.2 Å). Angles between subunits (S6E Fig) were determined by spline-interpolating points every 33.1 Å along the path of the pilus, as segmented in IMOD, calculating normal lines at each point, and determining the angles between the normal of each point.

## Bioinformatic analysis

Sequence data were downloaded from the Pseudomonas Genome Database version 20.2 [46] and filtered to exclude incomplete genomes. Searches were performed against every single strain using PSI-BLAST [74], with the CupA3 (PA2130 NCBI locus tag), CupB3 (PA4084), CupC3 (PA0994), and CupE5 (PA4652) proteins from the reference strain *P. aeruginosa* PAO1 as queries. Since the CupD system is missing in the strain PAO1, the CupD3 usher (PA14_59735) from strain UCBPP-PA14 was used as the query. To obtain unambiguous assignment of genes to CUP proteins, output data was further filtered with custom scripts and probable sequencing errors were corrected.

Structure predictions were performed with AlphaFold-Multimer version 2.1.1 [44,45], with sequences from the reference genome PAO1 as queries. Filaments were predicted without the signal peptide, which was predicted using SignalP [75]. The multiple sequence alignments (MSAs) used for the structure inference were built with the standard AlphaFold pipeline and the "reduced_dbs" preset. Template modelling was enabled and structures were inferred with eight MSA recycling iterations and all five different model parameter sets. After prediction, the models were ranked by the pTM score and only the highest-ranking model was selected. The PAE-value-plot for each structure is shown in S8 Fig. All predictions were performed using the high-performance computer "Raven", operated by the Max-Planck Computing & Data Facility in Garching, Munich, Germany. The multiple sequence alignment shown in S4 Fig was obtained by first calculating an initial alignment using PROMALS3D [76] in default settings and then curating it manually.

## Supporting information

**S1 Fig. Deletion of *cupA6* causes increased expression of CupE pili. (A)** Negative stain image of a *P. aeruginosa* PAO1 strain with *cupA6* deletion. Deletion of *cupA6* causes upregulation of cell-surface fibres compared to a control strain without the deletion (188 fibres in the Δ*cupA6* strain versus 60 fibres in a control strain; n = 30 micrographs; see Methods). Fibres

are highlighted by arrows. **(B-C)** Cell surface filaments were sheared, precipitated, and subjected to cryo-EM, showing (B) the Δ*cupA6* strain as shown in (A), and (C) a Δ*cupA6* Δ*cupE1-2* strain, demonstrating that the cell surface pili with a dotted zigzag pattern are CupE pili. Small fibre contamination is enriched in (C). This contaminant is possibly DNA due to its size and low persistence length, which precipitates under similar conditions [77]. **(D)** In the cryo-EM dataset of purified CupE pili, instances of CupE pili forming a crisscross mesh-like array were observed.
(TIF)

**S2 Fig. Cryo-EM of the CupE pilus. (A)** Atomic model of the CupE pilus, consisting of CupE1 subunits, in the transparent cryo-EM density at 15 σ away from the mean. **(B)** The same density shown in (A) coloured according to local resolution. **(C)** Independent half-map FSC curve. **(D)** Map-vs-model FSC curve.
(TIF)

**S3 Fig. Structural features of the CupE pilus. (A)** Hydrophobic surface depiction of an uncomplemented CupE1 subunit reveals that the donor strand inserts into a hydrophobic groove. **(B)** Straight and curved 2D class averages of CupE pili. Orange lines indicate the center of the filament to facilitate the visualization of curvature. Scale bars are 100 Å. **(C)** Top view of a five-subunit ribbon model of the CupE pilus reveals that a serine-threonine-rich loop (marked red, sequence TTTTSST) extends from the pilus, exposed to the environment. **(D-F)** Mutation of the first three residues of the loop shown in (C) to AGA ($111\text{-}113_{AGA}$) followed by structural determination at 4.1 Å resolution via cryo-EM shows reduced density near the loop, suggesting this density could arise from post-translational modifications. Density is shown at 15 σ contour level in (D) and (E), difference density is shown at 30 σ in (F). **(G)** Intact MALDI mass spectrometry of CupE preparations. Samples were treated 1:1 with 70% formic acid to trigger disassembly into monomers and spotted 1:1 with sinapinic acid. The major peak (16.126 kDa) corresponds to the predicted weight of CupE1 (16.122 kDa).
(TIF)

**S4 Fig. Multiple sequence alignment of major and minor pilin subunits of archaic CUP pili.** Conserved residues are shown in boldface. Secondary structure (b = β-strand, h = α-helix) is annotated based on our cryo-EM structure of the CupE1 filament. The conserved cysteine residues (C41 and C85) that form a disulfide bond in CupE1 filaments are highlighted in yellow and the serine-threonine-rich loop in cyan. The signal peptide in each sequence, as predicted using SignalP 6.0 [75], is underlined. Accession details for the shown protein sequences are provided in the S5 Table. Residues 62–69, which are located near the subunit-subunit interface, are marked in grey. L66 and V70, which interact with the donor strand within the same subunit fold, are conserved; residues facing the subunit-subunit interface are not conserved.
(TIF)

**S5 Fig. Comparison of CupE1 with other pilins. (A)** Comparison of the CupE1 subunit with CsuA/B (PDB 7ZL4, RMSD 1.545 Å) and PapA (PDB 5FLU, RMSD 5.21 Å). **(B)** Pilus architecture of the CupE pilus versus the Csu pilus. **(C)** Subunit contacts (<4 Å interaction) between main pilins within the CupE and Csu pilus. Interacting residues are marked in red. A loop contributing to the subunit interface in the Csu pilus, but not in the CupE pilus, is marked with an arrow.
(TIF)

**S6 Fig. CupE pili imaged on cells recapture features of isolated CupE pili. (A)** Cryo-ET of pili on cells (upper) recaptures the size and zigzag architecture of the atomic model of CupE, which was projected at 10 Å resolution for comparison (lower). Scale bar is 100 Å. **(B)** Subtomogram averaging of pili on cells results in zigzag-shaped density consistent with the atomic model shown as ribbons. Particles were aligned against a cylindrical reference to prevent bias. **(C-D)** Segmentation of tomograms as shown in Fig 3. **(E)** Quantification of local pilus curvature, by measuring angular deviation from the helical axis of the pilus on a *P. aeruginosa* cell (shown in Fig 3C).
(TIF)

**S7 Fig. Structural predictions of minor pilins CupE2 and CupE3 and the tip adhesin CupE6 by AlphaFold2. (A)** Predictions of homotetramers of CupE1, CupE2, and CupE3. The model for CupE1 was validated by comparison with the cryo-EM structure ($C_\alpha$-RMSD of E1/E1 subunits: 0.96 Å) (Figs 1 and 2). All pilins of CupE are predicted to share a similar structure ($C_\alpha$-RMSD E1/E2: 1.64 Å, E1/E3: 1.63 Å, E2/E3: 0.89 Å), and CupE2 and CupE3 are also predicted to polymerize through donor-strand complementation. Pilins mainly differ in domain orientation within the filament. **(B)** Predictions of heterotetramers consisting of two CupE1 and CupE2 subunits (upper) or two CupE1 and two CupE3 subunits (lower). The modelling suggests that donor strand exchange is possible between major and minor pilin subunits but does not reveal a preferred subunit arrangement. **(C)** Prediction of a filament consisting of CupE1 capped with the CupE6 adhesin tip subunit. The adhesin protein CupE6 consists of two domains, with the C-terminal domain capping the filament and the N-terminal domain exhibiting hydrophobic surface patches as predicted in previous studies [19]. **(D)** CupE6 adhesin domain prediction as in (B) shown as hydrophobic surface depiction, revealing a hydrophobic patch at the domain tip.
(TIF)

**S8 Fig. Prediction confidence measures for all tetramers shown in S7 Fig.** The prediction confidence is evaluated with different scores. Structures are colored according to their predicted Local Distance Difference Test (pLDDT). The pLDDT measures the model confidence per residue [44]. The pLDDT for all tetramers is very high or high for most parts of the prediction and low only for β-hairpin motifs (e.g. labeled as '1' in (A)). The PAE (Predicted Aligned Error) measures the expected positional error (in Å) at residue X, when the predicted and true structures are aligned on residue Y [44,78]. The PAE is visualized as a heatmap, where green means low expected error, showing the PAE for every pair of residues, resulting in 4x4 submatrices for every PAE heatmap. Every submatrix (e.g., labeled as '2') on the diagonal shows the intra-subunit PAE. It shows that the intra-domain PAE for all filaments and the PAE for the donor β-strand interaction (e.g., labeled as '3') is low, arguing that the protein structure for one filament subunit is predicted correctly. The inter-subunit predicted error increases when the distance between the subunits is greater. This implies on the one hand that the inter-subunit orientation prediction should not be taken as a precise measurement. On the other hand, it could mean that the filament is flexible and therefore no rigid inter-subunit conformation exists, consistent with our cryo-EM data. In (D), the relative domain position within the CupE6 adhesin is predicted with high confidence, but the PAE for the relative orientation of CupE6 against the CupE1 filament is high. This could again mean high flexibility, but also that the predicted adhesin orientation must be interpreted with care.
(TIF)

**S1 Table. CupE1 cryo-EM data acquisition and processing statistics.**
(DOCX)

**S2 Table. Primers used in this study.**
(DOCX)

**S3 Table. Plasmids used in this study.**
(DOCX)

**S4 Table. Strains used in this study.**
(DOCX)

**S5 Table. Accession details for proteins encoded by genes shown in Fig 4 and for proteins shown in S4 Fig.** For the complete assignment of NCBI gene loci to CUP gene clusters for every strain of *P. aeruginosa* of the *Pseudomonas* Genome Database, please refer to supplementary data file (S1 Data) additionally included with the manuscript.
(DOCX)

**S1 Data. Assignment of CUP clusters to *P. aeruginosa* genomes from the *Pseudomonas* genome database.** A "*" before the location in the supplementary file indicates that only a pseudogene was found. This is likely due to sequencing errors.
(XLSX)

**S1 Movie. Cryo-EM reconstruction of CupE1 from *P. aeruginosa*.** A 3.5 Å resolution cryo-EM density map of the CupE pilus is shown at an isosurface threshold of 15 σ away from the mean, which was used to build an atomic model of the main pilus-forming subunit CupE1. The zigzag-arranged Ig-like domains of CupE1 are complemented and stabilized by donor strand exchange between the individual subunits (surface depiction and ribbon diagrams shown).
(MP4)

**S2 Movie. Electron cryotomographic imaging of CupE pili on cells.** Tomographic 'Z'-slices through *P. aeruginosa* cells and their surrounding region are sequentially shown and reversed when the end of the pilus is reached. CupE pili emerge from the outer membrane of *P. aeruginosa* as extended flexible filaments, here appearing to attach to a particle within their surroundings.
(AVI)

## Acknowledgments

The authors would like to thank Dr. Thomas Clamens for help with strain generation.

## Author Contributions

**Conceptualization:** Alain Filloux, Tanmay A. M. Bharat.

**Data curation:** Jan Böhning, Nina Sulkowski, Andriko von Kügelgen, Abul K. Tarafder, Vikram Alva, Tanmay A. M. Bharat.

**Formal analysis:** Jan Böhning, Adrian W. Dobbelstein, Tanmay A. M. Bharat.

**Funding acquisition:** Alain Filloux, Tanmay A. M. Bharat.

**Investigation:** Jan Böhning, Nina Sulkowski, Sew-Yeu Peak-Chew, Mark Skehel, Tanmay A. M. Bharat.

**Methodology:** Vikram Alva.

**Project administration:** Tanmay A. M. Bharat.

**Resources:** Kira Eilers, Andriko von Kügelgen, Abul K. Tarafder, Vikram Alva.

**Software:** Adrian W. Dobbelstein, Vikram Alva.

**Supervision:** Alain Filloux, Tanmay A. M. Bharat.

**Validation:** Jan Böhning, Tanmay A. M. Bharat.

**Visualization:** Jan Böhning, Adrian W. Dobbelstein.

**Writing – original draft:** Jan Böhning, Tanmay A. M. Bharat.

**Writing – review & editing:** Jan Böhning, Adrian W. Dobbelstein, Nina Sulkowski, Kira Eilers, Andriko von Kügelgen, Abul K. Tarafder, Vikram Alva, Alain Filloux, Tanmay A. M. Bharat.

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
