## [Decision Letter · Decision Letter 0]

1 Sep 2022

Dear Dr Bharat,

Thank you very much for submitting your manuscript "Architecture of the biofilm-associated archaic CupE pilus from *Pseudomonas aeruginosa*" for consideration at *PLOS Pathogens*. As with all papers reviewed by the journal, your manuscript was reviewed by members of the editorial board and by several independent reviewers. The reviewers appreciated the attention to an important topic. Based on the reviews, we are likely to accept this manuscript for publication, providing that you modify the manuscript according to the review recommendations.

[1] A letter containing a detailed list of your responses to all review comments, and a description of the changes you have made in the manuscript. The same reviewers who provided comments for Review Commons were used for this submission to *PLOS Pathogens*. **In your responses, please also explain if and how the changes indicated in your Revision Plans for Review Commons were implemented.  **

Sincerely,

Matthew A Mulvey, Ph.D.

Associate Editor

PLOS Pathogens

Raphael Valdivia

Section Editor

PLOS Pathogens

Kasturi Haldar

Editor-in-Chief

PLOS Pathogens

orcid.org/0000-0001-5065-158X

Michael Malim

Editor-in-Chief

PLOS Pathogens

orcid.org/0000-0002-7699-2064

Reviewer Comments (if any, and for reference):

Reviewer's Responses to Questions

**Part I - Summary**

Reviewer #1: This is a relatively short manuscript, although well written and presented. The novelty lies in the study of archaic CUP pili, about which comparatively little is known. The main focus of the work lies in the cryoelectron microscopy and tomography studies of the CUP pili- this has been well executed and is a valuable contribution to the field. The technical challenges inherent in studying archaic CUP pili in P. aeruginosa are clear, requiring deletion of flagellum, type IV pili and upregulation of expression by deletion of a repressor. This is necessary for the tomographic reconstruction work but carries the obvious limitation that CUP behaviour may be influenced by this somewhat artificial environment (as acknowledged by the authors in the Discussion around line 320). The 3D structure of CupE1 appears to be very similar to other CUP pilins. A major point of interest lies in the zig-zag appearance of the pilus fiber, and the observations around its flexibility. The degree to which such flexibility can be associated with pilus function is debateable. There is a general absence of data which establishes this (for example, mutation to alter flexibility and demonstration that this has an effect on some measured biological function)

Reviewer #2: CupE pili are representatives of archaic CU pili which have a role in biofilm formation by several human pathogens.

This manuscript provides insights into the architecture of archaic CupE pilus from the opportunistic human pathogen Pseudomonas aeruginosa. For the revision of this manuscript, the authors plan to perform additional experiments to characterize the pili properties, which will further strengthen the manuscript. If executed, the manuscript will be suitable for PLOS Pathogens.

Reviewer #3: The authors report the cryoEM 3D reconstruction of the native P. aeruginosa CupE1 pilus, and use cryoET and subtomographic averaging to document the fiber architecture, global structure and positioning of CupE pili on the bacterial cell surface. CupE pili belong to the so-called archaic chaperone-usher pilus systems, found in important human and animal pathogens, and for which little structural information was available until recently. The report of the CupE1 pilus follows shortly on the recent report of the A. baumannii Csu pilus (Pakharukova et al. 2022), and appears to confirm the conservation of a zig-zag architecture in the main subunits of these archaic pilus systems. The structural biology work in this new manuscript is beautifully executed and forms a valuable addition to our understanding of these virulence structures and will be of interest to a broad audience.

**Part II – Major Issues: Key Experiments Required for Acceptance**

Reviewer #1: None required

Reviewer #2: In my previous comments I pointed out out minor issues. The authors are planning to address all of them.

Reviewer #3: In their new manuscript, the author respond to comments and suggestions I made in the context of an earlier submission. I am supportive of publication of the study, but remain of the opinion that the extend of flexibility in the CupE1 polymer is excessively emphasized. This is done in abstract (Ln 31-35), main text (Ln 216-218) and discussion (Ln 293-297; 316-319), and is suggested to be a functionally ‘key feature’ and to ‘contrast’ with the archaic Csu pilus of A. baumannii, which was recently described to assemble into a supereleastic zig-zag spring (Pakharukova et al. 2022). The reader is left with the image that CupE pili would be “akin a rope” that is flexible and wraps around other objects. I see no evidence for this strong emphasis. A couple of arguments to substantiate my opinion:

1. The cryoET data do show the presence of long range curvature in the CupE pili, but there is no real quantitative measure of the flexibility of the CupE1 polymer. Based on the images provided, the persistence length still looks quite large and the subtomogram averaging shows the fibers maintain the zig-zag architecture. Whilst this does show at least some flexibility in the subunit-subunit contact, there is no data to say this involves the GAG linker and the way now formulated may give the reader the impression that the contact between CupE1 subunits is essentially flexible, which is clearly not the case.

2. Looking at the presented structure, it appears the “clinch contact” shown to underlie zig-zag architecture and the superelastic nature of the Csu pilus is highly similar, if not conserved, in the CupE1 zig-zag architecture: i.e. Figure 2D the loop S62-V69 in subunit n and the pocket formed by G40 – W55 in subunit n+1.

A more detailed comparison between both structures should be included in the supplementary Figures and discussion.

Again, the subtomographic averaging shows this clinch contact remains essentially intact, but does allow long range flexibility in the fibers. It seems quite plausible to me that the spring-like and superelastic nature of Csu will proof to be conserved in CupE1, albeit with a lesser stiffness of the fiber.

3. The alphafold2 modelling of homopolymers of the different CupE subunits suggests that the potential clinch contact and resulting zig-zag architecture are unique to CupE1 (a feature conserved in the major subunits of archaic CUPs ?) and that the contacts zones between the minor CupE subunits (E2, E3 and E6) are less ordered. That would bring forward a picture where the zig-zag architecture in the archaic main pilus subunits is conserved, forming the equivalent of the more rigid helical packing in the rod of classical (i.e. gamma and pi clade) pilus systems, and that the minor subunits may form the equivalent of the more flexible tip fibrillae in classical pili, or main form hinge regions if incorporated within the polymer of the major subunit. The role of the minor subunits remains to be confirmed, but the authors could include potential scenarios in their discussion.

4. There is no data in the manuscript that allow any claim regarding the functional importance of the CupE1 curvature described in the manuscript. I think this should be discussed with more caution than is currently done.

**Part III – Minor Issues: Editorial and Data Presentation Modifications**

Reviewer #1: L155 'large mesh-like bodies'- Fig S1D; is this an artefact of concentration of purified pili or might it have some physiological significance?

L177 The curvature not very precisely described, given its importance to the manuscript. Can an angle be estimated?

L189 Did the mass spec analysis of CupE1 preparation indicate any evidence for glycosylation, as suggested in line 189

Reviewer #2: As mentioned in my previous comments, the manuscript needs a couple of sentences that clarify what is known and what is not known about subunit composition and subunit order of CupE pili.

Reviewer #3: - Ln 63. The authors reference the Hospenthal 2016 review for the donor strand complementation mechanism of subunit-subunit contacts. The primary reference for this principle is Sauer et al. 2002.

- In Ln180 – 194 the authors observed a residual density in a T/S rich loop that is suggestive of post-translational modification, likely O-glycosylation. Mutation of a TTT stretch to GAG results in loss of the residual density confirming the presence of the PTM. The assignment as a candidate glycosylation site seems plausible. Did the MS peptide fingerprint that was performed show the presence of the PTM?

PLOS authors have the option to publish the peer review history of their article (what does this mean?). If published, this will include your full peer review and any attached files.

Reviewer #1: **Yes: **Prof Jeremy Derrick

Reviewer #2: No

Reviewer #3: **Yes: **Han Remaut

Figure Files:

Data Requirements:

Reproducibility:

References:

---

## [Editor Report · Decision Letter 1]

3 Feb 2023

Dear Dr Bharat,

We are pleased to inform you that your manuscript 'Architecture of the biofilm-associated archaic Chaperone-Usher pilus CupE from Pseudomonas aeruginosa' has been provisionally accepted for publication in PLOS Pathogens.

Best regards,

Matthew A Mulvey, Ph.D.

Academic Editor

PLOS Pathogens

Raphael Valdivia

Section Editor

PLOS Pathogens

Kasturi Haldar

Editor-in-Chief

PLOS Pathogens

orcid.org/0000-0001-5065-158X

Michael Malim

Editor-in-Chief

PLOS Pathogens

orcid.org/0000-0002-7699-2064
---

## [Editor Report · Acceptance letter]

30 Mar 2023

Dear Dr Bharat,

We are delighted to inform you that your manuscript, "Architecture of the biofilm-associated archaic Chaperone-Usher pilus CupE from *Pseudomonas aeruginosa*," has been formally accepted for publication in PLOS Pathogens.

Best regards,

Kasturi Haldar

Editor-in-Chief

PLOS Pathogens

orcid.org/0000-0001-5065-158X

Michael Malim

Editor-in-Chief

PLOS Pathogens

orcid.org/0000-0002-7699-2064